# Intranasal Pregabalin Administration: A Review of the Literature and the Worldwide Spontaneous Reporting System of Adverse Drug Reactions

**DOI:** 10.3390/brainsci9110322

**Published:** 2019-11-13

**Authors:** Mohamed Elsayed, René Zeiss, Maximilian Gahr, Bernhard J. Connemann, Carlos Schönfeldt-Lecuona

**Affiliations:** Department of Psychiatry and Psychotherapy III, University of Ulm, Leimgrubenweg 12-14, 89075 Ulm, Germany; Rene.Zeiss@uni-ulm.de (R.Z.); Maximilian.Gahr@uni-ulm.de (M.G.); Bernhard.Connemann@uni-ulm.de (B.J.C.); Carlos.schoenfeldt@uni-ulm.de (C.S.-L.)

**Keywords:** pregabalin, misuse, abuse, dependence, snorting, sniffing, nasal administration

## Abstract

Background: It is repeatedly reported that pregabalin (PRG) and gabapentin feature a potential for abuse/misuse, predominantly in patients with former or active substance use disorder. The most common route of use is oral, though reports of sublingual, intravenous, rectal, and smoking administration also exist. A narrative review was performed to provide an overview of current knowledge about nasal PRG use. Methods: A narrative review of the currently available literature of nasal PRG use was performed by searching the MEDLINE, EMBASE, and Web of Science databases. The abstracts and articles identified were reviewed and examined for relevance. Secondly, a request regarding reports of cases of nasal PRG administration was performed in the worldwide spontaneous reporting system of adverse drug reactions of the European Medicines Agency (EMA, EudraVigilance database). Results: The literature search resulted in two reported cases of nasal PRG use. In the analysis of the EMA-database, 13 reported cases of nasal PRG use (11 male (two not specified), mean age of users = 34.2 years (four not specified)) were found. In two cases fatalities occurred related to PRG nasal use. Conclusions: Even if only little evidence can be found in current literature, the potential for misuse/abuse of PRG via nasal route might be of particular importance in the near future in PRG users who misuse it. Physicians should be aware of these alternative routes of administration.

## 1. Introduction

The gabapentinoids pregabalin (PRG) and gabapentin are widely used in primary healthcare, neurology, psychiatry, and for pain treatment [1]. Both substances are subject to misuse [2]. PRG is an alkylated analogue of gamma-aminobutyric acid (GABA) and binds to the 2 d type 1 protein of the P/Q voltage-dependent calcium channel and reduces the release of excitatory neurotransmitters in the central nervous system [3]. Its exact mechanism of action is not fully understood yet. In Europe, PRG is approved for the treatment of epilepsy (partial seizures), generalized anxiety disorder (GAD), and neuropathic pain. In the USA, PRG is not approved for the treatment of GAD, however, it is approved for the treatment of fibromyalgia and post-herpetic neuralgia [4]. It is also frequently prescribed off-label for bipolar disorder, alcohol/narcotic withdrawal, attention-deficit/hyperactivity disorder, restless legs syndrome, and trigeminal neuralgia [5]. PRG has also been used to treat anxiety in patients with schizophrenia [6,7,8]. Worldwide sales of PRG in 2014 reached about 5.4 billion USD, with an annual growth rate of about 12% [9]. Although PRG is generally considered a well-tolerated substance, euphoria as an adverse reaction has been reported in about 5% of the patients. There is an increasing number of reports revealing the potential of PRG for causing substance-use disorders [2,5,8,10,11,12]. In some of them, the PRG dosages used were up to 20 times higher than the recommended maximum dosage [13]. When considering substance-use disorders related to PRG, the main route of administration is the oral intake, but intravenous, rectal (‘plugging’), smoking and ‘parachuting’ (emptying the content of the capsule into a pouch) administration routes are being increasingly reported [14]. Higher PRG dosages administered intravenously or through mucosal tissue (nasal/lingual) may have other psychotropic effects than the ones already known and might be more dangerous, particularly when consumed in combination with other compounds, such as alcohol, gabapentin, benzodiazepines, cannabinoids, hallucinogenic substances (lysergic acid diethylamide (LSD) /Salvia divinorum), heroin/opiates, and amphetamines/synthetic cathinones as is the case in polytoxicomania [14]. The Swedish Poison Information Center reviewed all nationally registered cases regarding PRG related substance-use disorders from January 2011 to June 2013 and identified cases (number not specified) in which PRG was crushed and injected intravenously [15]. The time of onset of the psychotropic effects of PRG depends on the route of administration, which ranged from 10 minutes to two hours [14]. Reccoppa et al. reported that Florida inmates admitted snorting gabapentin powder for effects reminiscent of cocaine [16]. Although the phenomenon of intranasal administration of PRG might be very common in the consumer scene, this administration way appears to be receiving relatively little attention in the literature. Until now, to the best of our knowledge, only very few cases have been published (results file below [17,18]). This review aims to provide an overview of what is known from the literature and worldwide spontaneous reporting systems of adverse drug reactions regarding nasal PRG use.

## 2. Materials and Methods

We performed a narrative review of the currently available literature related to nasal use of PRG using the publicly available database MEDLINE (http://www.pubmed.com) of the National Library of Medicine (http://nlm.nih.gov), EMBASE, and Web of Science. One researcher (M.E.) performed the search in July 2018 using following search terms: “pregabalin” AND “nasal”, “intranasal”, “nasal administration”, “nasal consumption”, “nasal application”, “sniffing”, snorting”. Summaries of the retrieved abstracts and articles were reviewed and examined for relevance by two researchers (M.E. & C.S.-L.). A third party (R.Z.) was involved in case of uncertainty regarding the inclusion and evaluation of the articles. Furthermore, a request for reported cases from the worldwide spontaneous reporting system of adverse drug reactions Eudra Vigilance Data Analysis System (EVDAS), obtained from the European Medicines Agency (EMA) through BfArM (Federal German Ministry of Pharmaceutical Products and Medical Devices) was made (March 2018). The following search strategy was used for the search in EVDAS: “Substance” = “Pregabalin”; “way of administration” = “nasal”; “SMQ” = Drug abuse, dependence, and withdrawal (broad und narrow suspect, interacting, a drug not administered). Descriptive analysis of identified cases was conducted.

## 3. Results

### 3.1. Literature Search

The literature search in the MEDLINE, EMBASE, and Web of Science databases yielded two hits corresponding to two reported cases of nasal PRG administration (Ozturk et al. [17], and Snellgrove et al. [18]). Ozturk et al. in 2018 reported a 23-year-old female patient who presented with myoclonus and loss of consciousness following a high dose (not specified) of PRG applied via intranasal. The cranial CT was normal. A very short generalized spike-wave activity was detected in the electroencephalogram. The myoclonus resolved quickly after intravenous administration of valproic acid [17]. To detect PRG dependence or regular illicit use of it (misuse), Snellgrove et al. in 2017 carried out a one-year, cross-sectional study in 253 inpatients diagnosed with substance use disorder using a validated questionnaire and urine tests. More than half of the sample reported previous illicit use of PRG, 42% of which stated to have used at least one time PRG intra-nasal (parachuting) [18]. From the mentioned cohort, a 27-year-old man, who consumed PRG intranasally, suffered a severe intoxication (not otherwise specified), which was reported to the EVDAS (as shown in Table 1). 

### 3.2. Eudra Vigilance Data Analysis System (EVDAS) Search

On March 2018, a formal request for reported cases from the worldwide spontaneous reporting system of adverse drug reactions called “Eudra Vigilance Data Analysis System (EVDAS)”, obtained from the European Medicines Agency (EMA) through the national German Ministry “BfArM (Federal German Ministry of Pharmaceutical Products and Medical Devices)” was made. Using the above mentioned search strategy (see: Materials and Methods), 13 reported cases of nasal PRG application were retrieved. Four of the cases were retrieved from the UK, three from Germany, two from France, and one case each were reported in the following Countries: Austria, Denmark, Japan, and the USA, The first reported case dates from 2018 and the last registered from 2018. Only one case was retrieved from the literature (Snellgrove et al.); all the other cases were gained from the spontaneous reports. Nasal PRG consumers were predominantly male (11; in two of them gender not specified) and the mean age of users was 34.2 years (age in four not specified). In the majority of cases, there was no specific medical indication for PRG except for three cases (neuralgia, generalized anxiety disorder, and anxiety not otherwise specified). All reported cases had either current or past dependency or suffered from harmful use (here stated as intentional product misuse or abuse). In nearly half of the cases inpatient treatment was documented, but without any definite connection with the nasal application of PRG. In two cases fatalities occurred related to PRG nasal use. The summary of reported cases is shown in Table 1. 

## 4. Discussion

Regardless of the application form, the extent of PRG abuse/misuse is likely to increase in the coming years [11,19]. PRG can provoke a euphoric state [20], but higher dosages can also lead to a deep sedation or coma, particularly in combination with alcohol or other sedating drugs. Orally administered PRG is rapidly absorbed and reaches a maximum plasma concentration within one hour with an absolute bioavailability of at least 90%, irrespective of the dosage [20]. Pharmacokinetics of nasal or lingual PRG use have not been sufficiently investigated until now. The nasal or even the lingual administration of PRG powder contained in the capsules might be associated with a faster and more intensive psychotropic effect. Its quality (euphoric or sedative effect) will be mainly dependent on the person, but mostly on the dosage and the method of application used. Looking at the identified reported cases, male sex and history of substance abuse are major common characteristics for PRG abuse/misuse; this is in line with the existing literature [21,22,23,24,25,26,27]. Chiappini et al. performed a descriptive analysis of data retrieved from EudraVigilance database and observed that in 6.6% of all cases reporting PRG related adverse reactions (AR), the AR was due to abuse/misuse of the substance [28].

The data of the surveillance-databases of the EMA (EVDAS) showed 13 cases of nasal application of PRG. The cases listed here as a result from the search in the surveillance-databases of the EMA (EVDAS, Table 1) must be considered as “suspected cases” of adverse drug reactions related to the intranasal administration of PRG. This implies that a causal relationship cannot be established in each individual case. Furthermore, from spontaneous reports, it is not possible to estimate how often a specific undesirable effect occurs when using a medicinal product. In addition, information on how much more often a particular undesirable effect of one drug occurs in comparison to another drug is not available from such reports. The information derived from 13 reported cases between 2008 and 2018 has to be interpreted carefully, and its relevance should not be overestimated. From a numerical point of view, most cases (except for two cases: USA/Japan) come from Europe. This also raises the question of whether this type of application occurs especially in Europe. Interestingly, information about the different methods of PRG application can be retrieved from a drug forum from the website “Eve and Grave” and can be easily ordered directly through the website “Datmed” (personal information from users). Another point to consider and which should be subject to future research is the risks and adverse effects specific for nasal application. Until now there is not much knowledge if nasal application of pregabalin might increase the risk for damage to nasal passages, infections, or other adverse effects.

The rising trend of PRG abuse/misuse as reported by Snellgrove et al. shows that more than half of the inpatients treated because of a substance use disorder had an illicit concomitant current or former use of PRG [18]. The easy accessibility to PRG (by, e.g., ordering from the Web and not through legal medical prescription) might be a signal for a silent growing problem that should be urgently addressed. The relatively small number of publications regarding other ways of administration than the oral one also indicates a clear underrepresentation of this topic in literature. 

## 5. Conclusions

The intranasal PRG application might be much more frequent as assumed by deriving the data from spontaneous reports of surveillance databases or from current literature in medical databases, and might lead to more complications. Physicians should pay attention to other administration methods of PRG, especially in patients with a history of substance misuse/abuse. Healthcare professionals are encouraged to report any cases of PRG use outside the medical prescription to either the marketing authorization holder or the national pharmacovigilance surveillance programs of each country. 

## Figures and Tables

**Table 1 brainsci-09-00322-t001:** Enhanced Individual Case Line Listing provided from Eudra Vigilance Data Analysis System (EVDAS), adapted.

Country	Receive Date	Age, Sex	Primary Source Qualification	Serious	Death	Hosp.	Literature Reference	Indication	Concomitant Drugs	Reaction List/Outcome/Medical History
**USA**	10 December 2008	NS	Healthcare pro	No	No	No	No	No	No reported	intentional misuse/somnolence/ NA
UK	03 Janauary 2013	28 male	Healthcare pro	Yes	No	Yes	No	No	methadone	drug abuse/syncope, recovered/NA
UK	26 July 2013	31 male	Healthcare pro	Yes	No	No	No	No	methadone, DZP	drug dependence/unknown/NA
Den	23 February 2012	20 male	Healthcare pro	Yes	No	Yes	No	GAD	ZLP, Que, Xep	drug abuse/seizure, limb injury, not recovered/schizophrenia
Japan	22 August 2014	63 male	Healthcare pro	Yes	Yes	No	No	neuralgia	nedaplatin, FRS, PCT	drug-induced liver injury/intentional misuse/death/oesophageal carcinoma/DM-II
UK	29 October 2014	NS male	Healthcare pro	Yes	Yes	No	No	No	No reported	death/NA
Ger	11 September 2014	42 male	Healthcare pro	Yes	No	Yes	No	No	No reported	drug dependence/withdrawal/unknown/hepatitis C/drug dependence
France	29 November 2016	35 male	Healthcare pro	Yes	No	Yes	No	No	clonazepam, PCT	intentional product misuse/recovered/Asthma/drug dependence
France	13 December 2016	35 male	Healthcare pro	Yes	No	Yes	No	No	No reported	conduction disorder/drug dependence/unknown/drug dependence/subutex
UK	23 June 2017	27 male	Non-Healthcare pro	Yes	No	No	No	anxiety	No reported	anxiety/intentional misuse/condition aggravated/NA
Ger	08 November 2017	NS	Healthcare pro	Yes	No	No	No	No	No reported	drug abuse/unknown/NA
Ger	18 January 2018	27 male	Healthcare pro	No	NA	NA	Snellgrove et al.	No	No reported	drug abuse/unknown/NA
Austria	25 January 2018	NS male	Non-Healthcare pro	Yes	No	No	No	No	No reported	drug abuse/euphoric mood/unknown/ drug withdrawal maintenance

NS = not specified; AA = not available; SAR = severe adverse reaction; Hosp. = Hospitalization; GAD = generalized anxiety disorder; DM = diabetes mellitus; DZP = Diazepam; PCT = Paracetamol; ZLP =Zolpidem; Que = Quetiapine; Xep = Xeplion (Paliperidone Palmitate); FRS = Furosemide; PCT = Paclitaxel; NA = non-applicable; Ger = Germany; Den = Denmark.

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
