# Peer review of "Intranasal Pregabalin Administration: A Review of the Literature and the Worldwide Spontaneous Reporting System of Adverse Drug Reactions"

_brainsci, 2019, doi:10.3390/brainsci9110322_

Round 1

Reviewer 1 Report

This is an important review. 

I have only one comment. When giving the few cases from the search in the surveillance-databases reports, it would be important to add the number of such cases by other way of administration,  to see how rare (or not) the nasal way as compared to others.

Reviewer 2 Report

Thank you for allowing me to review this article that sought to identify cases of PRG administered intranasally via MEDLINE search and via query of EVDAS. While important, I feel the manuscript loses strength in the results/discussion section as the identified cases/reports are not fully described in results (only 2 from MEDLINE are discussed, the rest are in a table which is fine given the limited information but there needs to be context to support the table). In the discussion, the authors reiterate what was found, but do not link to why healthcare professionals should be aware of this use, or what the other dangers may be? Also, what do we do when we identify someone is using PRG intranasally? Do we engage in a slow taper of oral formulation? There are many questions left to be answered from this report. Lastly, this paper is missing a large portion of available literature regarding gabapentinoid misuse, particularly from the USA.

Abstract:

-Overall: avoid first person language such as “we”

-Page 1, line 19: Pregabalin and gabapentin are still being reported for this, recommend removing past tense language to reflect ongoing issue

-Page 1, line 20: Change to “predominantly in former or active substance use disorder” (i.e. active rather than actual since former diagnoses were also actual diagnoses; remove hyphen between substance and use as it is not hyphenated in DSM-5)

-Page 1, line 21: Merge and change 2 sentences to avoid using stigmatizing language and/or street terms in abstract; recommend “The most common route of use is oral, though reports of sublingual, intravenous, rectal, and smoking administration also exist.” Can keep street terms in introduction because they are explained.

-Page 1, line 23: Recommend changing consumption to use in this sentence, and all other sentences

-Page 1, line 24: Change “to” to “of”

-Page 1, lines 31-34: Recommend changing to “The potential for misuse/abuse of PRG via nasal route might become an increasingly important issue in our society”

Introduction:

-Page 1, line 39: Are you trying to say that these substances have an increasing prevalence of misuse? If so, consider revising because the drugs themselves are not “causing” misuse, but rather, they are subject to misuse.

-Page 1, line 44: Be consistent with abbreviating pregabalin, use PRG.

-Page 1, line 45: Licensed may not be the right word. Personally, I would stick with using the word “approved”

-Page 1, line 48: Change to “patients with schizophrenia” to avoid stigmatizing language and avoid describing people by their medical illness

-Page 1, line 63: Expand on the onset sentence – which routes provide quicker onset? What is the anticipated onset via intranasal route? Why are we medically concerned about intranasal route? Is it corrosive or mucosal membranes?

-Page 1, line 68: Either choose one or two, but not one two.

Materials & Methods:

-Page 2, line 74: EMBASE and Web of Science should not be a standalone sentence.

-Page 2, line 75: How many researchers performed the search?

-Page 2, line 78: How were disagreements amongst 2 researchers resolved?

-Page 2, line 84: This sentence is half methods, half results. Recommend changing to “Descriptive analysis of patient cases was conducted”. Then can put “Summary of patient cases can be found in Table 1” somewhere in the results

Results:

-Need to discuss all 13 cases, not just 2.

-Need to discuss all medical outcomes, including those who died

Discussion:

-Page 5, line 112: Recommend changing to “..is likely to increase in the upcoming years”

-Page 5, line 121: Change mayor to major

-Page 5, line 121: Cannot conclude that these are risk factors, but you can conclude that these are common characteristics. A risk factor must increase the risk of something developing, and you do not have evidence to support that.

-Page 5, line 121: You’re missing a lot of literature supporting these claims and others; see authors Peckham AM (Boston Massachusetts USA); Evoy KE (Texas USA); Gomes T (Ontario Canada); Vickers-Smith R (Kentucky USA)

-Overall, need to expand on the estimated risk of someone using PRG intranasally outside of euphoria and death. Damage to nasal passages? Increased risk for infection? Increased risk for cancer? Please discuss.
